# Broadening of the ν_2_ Raman Band of CH_4_ by C_3_H_8_ and C_4_H_10_

**DOI:** 10.3390/molecules28083365

**Published:** 2023-04-11

**Authors:** Aleksandr S. Tanichev, Dmitry V. Petrov

**Affiliations:** 1Institute of Monitoring of Climatic and Ecological Systems, Siberian Branch of the Russian Academy of Sciences, 634055 Tomsk, Russia; dpetrov@imces.ru; 2Department of Optics and Spectroscopy, Tomsk State University, 634050 Tomsk, Russia

**Keywords:** Raman spectroscopy, methane, propane, n-butane, isobutane, broadening coefficients, pressure, alkanes

## Abstract

Raman spectroscopy is a promising method for the analysis of natural gas. It is necessary to account for the broadening effects on spectral lines to improve measurement accuracy. In this study, the broadening coefficients for methane lines in the region of the ν_2_ band perturbed by propane, n-butane, and isobutane at room temperature were measured. We estimated the measurement errors of the concentration of oxygen and carbon dioxide in the case of neglecting the broadening effects on the methane spectrum by the pressure of C_2_–C_6_ alkanes. The obtained data are suited for the correct simulation of the methane spectrum in the hydrocarbon-bearing gases and can be used to improve the accuracy of the analysis of natural gas by Raman spectroscopy.

## 1. Introduction

Raman spectroscopy has been actively developed over the last decade as a tool for the analysis of various gaseous media. The appearance of commercially available powerful small-sized lasers and highly sensitive photodetectors on the market facilitated a significant amplification in the signal intensity. One of the most promising application fields for Raman spectroscopy is the analysis of methane-bearing fuel gases [1,2,3,4,5,6,7,8,9,10,11] since a large number of different compounds can be measured simultaneously by this technique using a single non-tunable laser. According to recent studies [4,5,12,13,14,15], Raman spectrometers have reached a limit of detection (LOD) below 100 ppm, which is close to the standards for gas chromatography [16]. In practice, it is necessary to take into account features of spectrum formation of the analyzed components to achieve high measurement accuracy in a wide range of concentrations. Characteristic Raman bands are sensitive to variations in the density and composition of a gas sample [17,18,19,20,21,22,23] due to the effects of intermolecular interaction. Thus, the resulting changes in a spectrum caused by physical processes will lead to a measurement error, despite the high sensitivity of a spectrometer [24,25]. The study of vibrational bands of methane is a priority task since they are the most intense in the spectrum of methane-bearing fuel.

The required spectral characteristics can be accurately simulated using quantum chemical calculations based on the tensorial formalism and theory of rotational-vibrational relaxation [26,27,28]. A good agreement between experimental and simulated Raman spectra of pure methane in a wide range of gas density was demonstrated earlier [29,30,31]. In addition to pure gases, this method is also suitable for simulating the influence of different compounds on the spectrum. Several experimental studies of mixtures of methane highly diluted in nitrogen, hydrogen, argon, and helium [32,33,34,35,36,37,38] showed good agreement with the model’s observations. Broadening coefficients of the spectral lines are required to calculate a spectrum in the case of variations in the pressure and composition of a gas mixture. At the moment, these parameters have been experimentally measured or evaluated for the following types of intermolecular interaction: CH_4_–CH_4_ [39,40,41,42,43,44], CH_4_–N_2_ [45,46,47], CH_4_–O_2_ [48,49,50], CH_4_–CO_2_ [51,52,53], CH_4_–He [46,50,54], CH_4_–Ar [55,56,57], CH_4_–H_2_ [58,59,60,61], and CH_4_–C_2_H_6_ [18]. Propane, n-butane, and isobutane are important components of natural gas, a valuable fossil fuel. It is necessary to study the effect of these gases to adequately simulate methane bands in such an environment. However, any measurements or theoretical evaluation of the corresponding coefficients have not previously been reported. In this study, we determined the *n*-C_4_H_10_-, *i*-C_4_H_10_-, and C_3_H_8_-broadening coefficients of methane lines in the region of the ν_2_ band using a high-sensitivity spectrometer based on spontaneous Raman scattering.

## 2. Results and Discussion

### 2.1. Methane Lines Broadened by Propane and Butanes

The structure of a methane spectrum is quite complex due to the high symmetry of the molecule. The four fundamental modes ν_1_ (2917 cm^−1^), ν_2_ (1533 cm^−1^), ν_3_ (3020 cm^−1^), and ν_4_ (1311 cm^−1^) exhibit an approximate relation of stretching and bending vibrations (e.g., with ν_1_ ≈ ν_3_ ≈ 2ν_2_ ≈ 2ν_4_ ≈ ν_2_ + ν_4_ in pentad) resulting in vibrational levels being grouped into clusters with levels of similar energy, called polyads [62,63]. The first is a dyad and includes the ν_2_ and ν_4_ vibrational modes, covering a wide range of 800–2000 cm^−1^ at room temperature [30]. The very weak transitions of the normally Raman inactive ν_4_ band appear in the Raman due to the Coriolis interaction with the active ν_2_ band [64,65] and can be neglected in the studied spectral range. Figure 1 and Figure 2 show the methane spectra in the 1505–1825 cm^−1^ range obtained by subtracting the contribution of propane and butanes. This region includes the rotational-vibrational lines belonging to the Q-branch (transitions with Δ*J* = 0), R-branch (Δ*J* = +1), and S-branch (Δ*J* = +2). About 700 lines in the range of 1533–1560 cm^−1^ form the broad Q-branch. Its rotational structure cannot be resolved using the described Raman spectrometer since these lines are very close to each other. Two series of equidistant peaks in the range of 1550–1825 cm^−1^ belong to the low-intensity R-branch and intense S-branch. Rotational constant *B*_0_ is 5.241 cm^−1^ for methane molecules [26,66]. Therefore, the wavenumber separation between the lines sharing the levels *J* and *J* + Δ*J* is about 10.5 cm^−1^ in the R-branch and about 21 cm^−1^ in the S-branch. Determination of the broadening coefficients in the range of 1550–1700 cm^−1^ is a bit more complicated since the R-lines sharing even-*J* levels partially overlap the S-lines. In this paper, we measured the methane lines up to *J* = 13 since the higher transitions are difficult to distinguish in a spectrum due to their low population at room temperature.

As shown in Figure 1, the peaks of the Q-, R-, and S-branches are broader in the spectra of binary mixtures compared to that of pure methane. The difference between the broadening effects of methane by alkanes C_3_–C_4_ is seen in the strong lines. At the same mixture pressure, the smallest broadening is observed in the mixture with propane and the largest with isobutane. This deviation becomes more pronounced with increasing pressure, confirming that these effects are caused by the collisional broadening of the methane lines. The obtained spectra were fitted using a Raman spectra simulation algorithm to retrieve the broadening coefficients. The model spectrum was computed as a convolution of a molecular spectrum and an instrument response function (IRF). The molecular spectrum was calculated as the sum of the profiles of rotational-vibrational lines. The line profile was assumed to be a Lorentzian distribution since Doppler broadening effect at room temperature is negligible relative to collision broadening at pressures above 0.1 MPa. The positions and intensities of methane lines at 296 K in the range of 1505–1825 cm^−1^ were taken from quantum calculations [67]. In the case of a binary gas mixture CH_4_/X, the width and position of the Lorentzian profile are defined as [68]:(1)$$\Gamma (P)=P\left(C_{CH_{4}}\gamma_{CH_{4}-CH_{4}}+(1-C_{CH_{4}})\gamma_{CH_{4}-X}\right),$$
(2)$$\nu (P)=\nu_{0}+P\left(C_{CH_{4}}\delta_{CH_{4}-CH_{4}}+(1-C_{CH_{4}})\delta_{CH_{4}-X}\right)$$
where Γ(*P*) and *ν*(*P*) are the half-width at half-maximum and position of the methane line at the total pressure *P*, *ν*_0_ is the line position in vacuum, $C_{CH_{4}}$ is the methane mole fraction in the mixture, $\gamma_{CH_{4}-CH_{4}}$ and $\delta_{CH_{4}-CH_{4}}$ are the self-broadening and self-shift coefficients of the line, and $\gamma_{CH_{4}-X}$ and $\delta_{CH_{4}-X}$ are the broadening and shift coefficients of the line perturbed by component *X*, respectively. The coefficients *γ* and *δ* were variable parameters of the model and were optimized by applying a multispectrum fitting procedure to all recorded spectra. The number of allowed transitions from each level *J* is proportional to *J*(*J* + 1) due to the tetrahedral splitting [67,69,70]. The observed peaks R(*J*) and S(*J*) in the recorded spectra are manifolds whose fine rotational structure is masked by collisional and instrumental broadening effects. In this regard, we determined the average broadening coefficients as a function of *J*. For this purpose, the parameters *γ* and *δ* were constrained to be identical for the lines inside each R(*J*) and S(*J*) manifold. As a result, 50 independent parameters were fitted using the least-squares fitting routine. Initial values of the broadening parameters $\gamma_{CH_{4}-X}$ and $\delta_{CH_{4}-X}$ were taken from Ref. [43]. It was found that the measured and calculated spectra agree within 4% using the retrieved parameters *γ* and *δ*. An example of the fitting results for the mixtures CH_4_/C_3_H_8_ and CH_4_/*n*-C_4_H_10_ is shown in Figure 2.

The measured broadening coefficients for the R- and S-branch are summarized in Table 1. It was found that the methane–propane, methane–n-butane, and methane–isobutane broadening coefficients are 38%, 45%, and 50% higher than the self-broadening coefficients, respectively. The shift coefficients of methane lines perturbed by these alkanes were found to be −0.075 ± 0.03 cm^−1^/MPa and cannot be clarified from the obtained spectra. The error of the fitted coefficients for the R(1, 2, 10–13) and S(0, 11) lines is higher than the other transitions because these lines are less intense.

### 2.2. Evaluation of Broadening Effect by Heavier Alkanes

The pressure broadening and shift coefficients of spectral lines depend on the interaction potential of two colliding particles [71,72]. A simplified approach was proposed by Zameroski et al. [73] to calculate the collisional broadening coefficients for the rubidium absorption line in the medium of C_1_–C_4_ alkanes. The basic idea of the model is to account only for the first nonzero term in the electrostatic multipole expansion of potential for two unlike interacting atoms (or molecules), which is related to the van der Waals attraction forces. Within this approximation, the broadening coefficients are proportional to *α*^0.4^*µ*^−0.3^, where *α* is the polarizability of the buffer gas particle, and *µ* is the reduced mass of colliding pair of particles (buffer-active). A good agreement between the measured and calculated broadening coefficients was demonstrated in Ref. [73]. We suppose that the nature of the methane–molecule interaction at medium and long ranges is similar to an atom–molecule interaction due to the spherical symmetry of tetrahedral molecules. Therefore, the purely attractive van der Waals potential is suitable for predicting the collisional broadening coefficients of methane. Experimentally determined broadening coefficients of methane lines perturbed by various gases were taken from refs. [18,39,46,48,49,50,51,52,53,54,57,74,75,76,77,78] to test this hypothesis. The polarizabilities of these atoms and molecules were taken from the extensive study by Gussoni et al. [79]. It is worth noting that the average value of the broadening coefficients decreases with increasing both quantum number *J* and the wavenumber center of a vibrational band [80,81]. In this regard, we used the $\gamma_{CH_{4}-X}/\gamma_{CH_{4}-CH_{4}}$ ratio averaged over all lines to improve the flexibility of this approach.

As shown in Figure 3, the obtained dependence of the broadening coefficients on the product *α*^0.4^*µ*^−0.3^ is very close to the linear for both low- and high-polarizability particles. The exception is the hydrogen–broadening coefficient, which is overestimated within this approximation. We estimated the ratios of the broadening coefficients of methane lines perturbed by other compounds that were not measured earlier. The results are summarized in Table 2. As expected, heavier hydrocarbons amplify the effect of collisional broadening. The broadening coefficients methane–pentane and methane–hexane are 60% and 71% higher than the methane–methane coefficients, respectively. Despite the proximity of the polarizabilities of n-butane and isobutane, the measured broadening coefficients of methane–n-butane and methane–isobutane differ by 5%. We suppose that the collisions with butanes, which cause perturbation of the methane molecule, occur at a sufficiently close range when a difference in the arrangement of their methyl groups begins to show. Most likely, a similar close-range interaction effect occurs when pentane and hexane isomers collide with methane. In this regard, the estimated values for C_4_-, C_5_-, and C_6_-broadening coefficients given in Table 2 should be considered averaged values over all isomers.

### 2.3. Influence of Methane Broadening by C_2_–C_6_ Alkanes on Measurement Error

Let us estimate the effect of the interaction between methane and heavier alkanes on the measurement error of the natural gas components by Raman spectroscopy. The environment of alkanes will lead to a measurement error of the species whose characteristic peaks are in the region of the ν_2_ band of methane. First, these include the Fermi dyad of carbon dioxide and the fundamental vibrational mode of oxygen (see Figure 4). The characteristic oxygen peak is formed by the Q-branch lines occupying the range of 1550–1557 cm^−1^. The effect of methane broadening on the measurement error of oxygen is significant because its peak overlapped by both the R(1) line and the high-intensity wing of the Q-branch of methane. Carbon dioxide bands are less sensitive to changes in the methane spectrum. The characteristic peak centered at 1388 cm^−1^ is located between the O(7) and O(8) transitions of methane, where the intensity variation is much smaller in comparison with the oxygen region.

The measurement error can be quantified by the following expression:(3)$$Error=\frac{\left|A_{CH4}^{*}-A_{CH4}\right|}{A_{X}}\times 100\%,$$
where $A_{X}$ is the integrated intensity of the Q-branch of O_2_ or CO_2_ in a pure gas, $A_{CH4}^{*}$ and $A_{CH4}$ are the integrated intensities of pure methane and broadened methane by heavier alkane at the same gas pressure, calculated in the range of the characteristic peak (1543–1563 cm^−1^ for O_2_ and 1384–1393 cm^−1^ for CO_2_). We used spectrum simulation to calculate the intensities in the specified ranges. The computation algorithm was the same as described in Section 2.1. The IRF of a spectrometer was imitated by a Gaussian distribution. Two sets of spectra with different IRF (the Gaussian widths were set to 0.5 and 5 cm^−1^) were simulated to study the effect of instrumental broadening. According to Equation (1), the greater the concentration of the perturbing component in the mixture, the more the line width of methane changes at a fixed pressure. Therefore, neglecting the collisional broadening methane–alkane will result in a large measurement error for the natural gas sample with a high content of hydrocarbons. In this regard, the maximum concentrations for simulating the spectra were chosen according to the upper limit of the molar fraction range of alkanes in natural gas [82]: 15% for ethane, 5% for propane, 2% for butanes, 1% for pentanes, and 0.5% for hexanes.

As expected, the measurement error of oxygen and carbon dioxide increases with increasing pressure at a fixed concentration of alkanes (see Figure 5). The decrease in the error at pressures above 5 MPa is related to the model features used to simulate the methane spectrum. The intensity of the Lorentzian wings spreads significantly from the center of a line at high density. This redistribution reduces the integrated intensities in a given range and their difference. The influence of ethane and propane on the ν_2_ band of methane is the most significant in natural gas since their concentration is the highest among hydrocarbon impurities. According to ISO 6974-6 [83], the LODs of oxygen and carbon dioxide in natural gas are 70 and 10 ppm, respectively. It can be seen that the measurement error in the case of neglecting the C_2_- and C_3_-broadening exceeds the LOD already at pressures above 0.5 MPa. The broadening of the ν_2_ band by butanes is less critical and can be neglected at pressures up to 2 MPa. The measurement error caused by the influence of heavier alkanes C_5_–C_6_ is within the LOD of O_2_ and CO_2_ in the pressure range of 0–10 MPa. The obtained curves demonstrate that the measurement errors are smaller when using a low-resolution spectrometer. The influence of different molecular environments on a spectral band is more masked by the wide IRF due to a smaller relative change in line widths in the molecular spectrum.

## 3. Methods

An experimental setup based on a 90° scattering geometry (see Figure 6) was used to record Raman spectra. Polarized radiation from a 532-nm non-tunable solid-state continuous-wave Nd:YAG laser (SLN-532-5000, Cnilaser, Changchun, China) with an output power of 5 W was focused into a metal gas cell to excite scattering. The cell was equipped with two entrance windows and one exit window. The volume of the sample chamber was about 30 cm^3^. The gas temperature was monitored using the internal thermocouple and maintained at 298 ± 1 K. A multipass excitation system provided the necessary sensitivity in the region of low-intensity methane lines. The multipass cavity was made of two spherical, highly reflective mirrors with a separation of about 200 mm. The laser beam is reflected between the mirrors through the entrance windows of the gas cell [84]. The scattered light was collected by two objective lenses: a collecting f/4 lens (f = 105 mm) and a focusing f/4.5 lens (f = 210 mm). Rayleigh scattering was removed by a holographic notch filter placed between the lenses. The collected light was focused into an f/8-spectrometer equipped with a 0.03 × 4 mm entrance slit, a 2400 lines/mm grating, and a Hamamatsu S10141 CCD camera (2048 × 512 pixels) with thermoelectric cooling to −10 °C. The simultaneously recorded spectral range was 1505–1825 cm^−1^ with a resolution of 0.5 cm^−1^ at 1600 cm^−1^. The relationship between the CCD pixels and wavenumbers was obtained using a third-order polynomial to transform an intensity distribution of decomposed light into a spectrum. We determined the polynomial coefficients by matching lines in the experimental spectrum of pure methane at atmospheric pressure to the simulated ones. The error of the obtained wavenumber scale was found to be ±0.005 cm^−1^.

On the one hand, the fraction of a foreign-gas in a mixture with methane should be high enough to detect the effect of a foreign-gas broadening the methane lines. On the other hand, the number density of methane must be high to provide a sufficient Raman signal. According to the criterion proposed by J. Kojima and Q.-V. Nguyen [85], the maximum pressure of a methane mixture when the impact approximation is valid can be estimated as:(4)$$P_{\max}=\frac{\nu_{sep}}{\Gamma },$$
where *ν_sep_* is the averaged line separation between consecutive rotational-vibrational lines of methane, and Γ is the average width (fwhm) of methane lines (was set to 1.4 cm^−1^/MPa). Using positions of methane lines from Ref. [67], values for *ν_sep_* and *P*_max_, averaged over three spin symmetries of methane, were found to be about 0.7 cm^−1^ and 0.5 MPa, respectively. Thus, we recorded the spectra of binary mixtures of methane/propane, methane/n-butane, and methane/isobutane (60/40 mol%) at pressures between 0.1 and 0.44 MPa to study the broadening processes. We believe the chosen mole fraction is adequate to provide sufficient signal-to-noise and the measurable effect of C_3_–C_4_ alkanes. The mixtures were prepared in a separate mixing chamber using pure samples of methane (>99.99%), propane (>99.8%), n-butane (>99.75%), and isobutane (>99.5%). The measurement error of the mole fraction in the obtained mixtures was estimated to be within 1%. Pressure in the gas cell and the mixing chamber was measured with 0.1% accuracy using a pressure gauge. It should be noted that the broad bands of CH_3_ deformation and CH_2_ scissor vibrational modes of the studied alkanes are located in the region of the ν_2_ band of methane [86]. In this regard, we also recorded the spectra of pure propane and butanes to extract the methane band from the spectra of mixtures.

## 4. Conclusions

In this work, the averaged broadening coefficients for the R(*J*) and S(*J*) manifolds of the ν_2_ band of methane perturbed by propane and butanes were measured for the first time. It was found that the methane–propane and methane–butanes interactions significantly amplify the collisional broadening of methane lines in comparison with the methane–methane broadening. The C_3_H_8_-, *n*-C_4_H_10_-, and *i*-C_4_H_10_-broadening coefficients were found to be 38%, 45%, and 50% higher than the CH_4_-broadening, respectively. Foreign-gas broadening parameters for the R(1–10) and S(0–10) transitions were measured within a precision of 7% based on uncertainties from mole fraction, gas pressure, signal-to-noise of the recorded spectra, and instrument response function. Further improvement of the measurement accuracy of the broadening coefficients is possible mainly by enhancing the spectral resolution. The effect of foreign-gas broadening will be more pronounced in those spectra where the contribution of the instrument response function will be less. Higher resolution and signal intensity can be achieved by using nonlinear Raman spectroscopy, e.g., coherent anti-Stokes Raman scattering or stimulated inverse Raman scattering. The methane spectrum can be correctly simulated using the line list for methane and obtained parameters. We believe that our data will improve the accuracy of the analysis of methane-bearing gases by Raman spectroscopy.

We have shown that the van der Waals potential, which accounts for only attractive forces, is a suitable approximation to predict the collisional broadening coefficients for rotational-vibrational lines of methane in various media. Within this approximation, the broadening coefficients are proportional to the product of the polarizability of a perturbing molecule and the reduced mass of colliding molecules. We have estimated the averaged C_5_H_12_-, C_6_H_12_-, CO-, and H_2_S- broadening coefficients of methane lines from this relationship for the first time. The broadening parameters for *J*-manifolds can be reconstructed from the distribution of methane self-broadening coefficients by quantum number *J* and the provided ratios. We suppose that this model is also suitable for other tetrahedral molecules. The necessary broadening coefficients for the active compound can be estimated from the polarizabilities of perturbing molecules or atoms.

Absolute measurement error of oxygen and carbon dioxide in natural gas increases with increasing gas density in the case of neglecting different broadening effects on the ν_2_ band of methane. If the O_2_ or CO_2_ content is close to the limit of detection, the effect of ethane, propane, and butanes on the measurement error of O_2_ and CO_2_ cannot be neglected at gas pressures above 0.5, 1, and 2 MPa, respectively. These measurement errors will also result in a deviation of the measured concentrations of all other components from the actual values. Heavy alkanes (pentane, hexanes, and heavier) slightly broaden the methane spectrum due to their low content in natural gas. The measurement errors of O_2_ and CO_2_ do not exceed the limit of detection in this case.

## Figures and Tables

**Figure 1 molecules-28-03365-f001:**
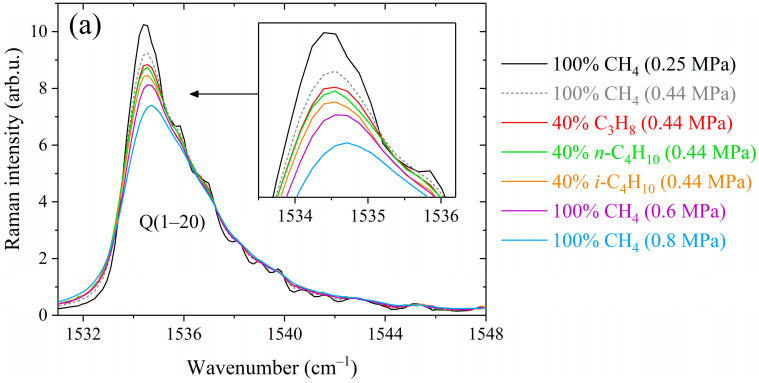
Comparison between methane Raman spectra recorded in pure methane and binary mixtures containing 60/40 mol% methane/propane, methane/n-butane, and methane/isobutane at different pressures in the (**a**) Q-branch region, (**b**) R(3) manifold, (**c**) R(10) and S(4) transitions, (**d**) S(6) manifold, and (**e**) S(9) manifold. The contributions of alkanes to the spectra were removed. All spectra are normalized by the integrated intensity.

**Figure 2 molecules-28-03365-f002:**
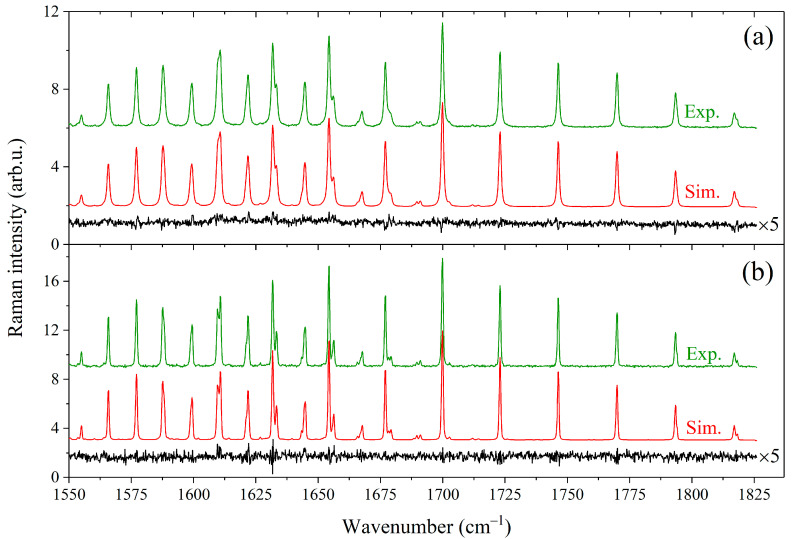
Comparison between experimental spectra and simulations generated using the fitted broadening coefficients. (**a**) Spectra of the 60/40 mol% CH_4_/C_3_H_8_ mixture at a pressure of 0.44 MPa. (**b**) Spectra of the 60/40 mol% CH_4_/*n*-C_4_H_10_ mixture at 0.1 MPa. The contributions of alkanes to the spectra were removed. All spectra are normalized by the integrated intensity. The residual (experimental minus simulated) is plotted at the bottom of each spectrum (multiplied by 5).

**Figure 3 molecules-28-03365-f003:**
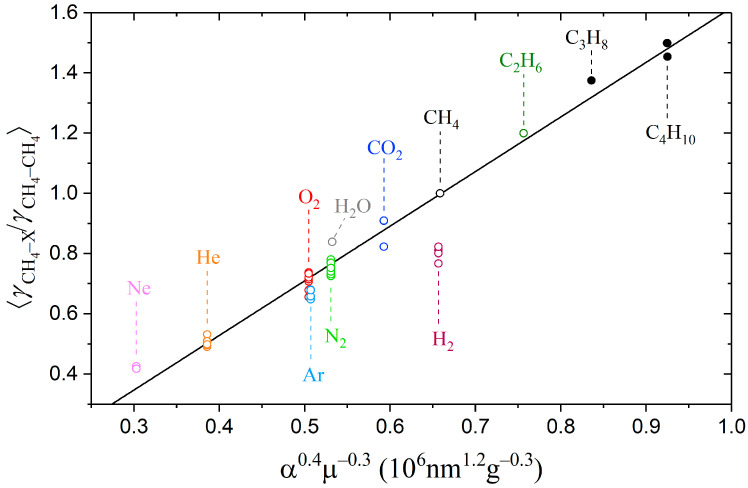
Averaged ratios of the broadening coefficients $\langle \gamma_{CH_{4}-X}/\gamma_{CH_{4}-CH_{4}}\rangle$ plotted as a function of *α*^0.4^*µ*^−0.3^, where open circles ○ denote experimental data taken from refs. [18,39,46,48,49,50,51,52,53,54,57,74,75,76,77,78] and closed circles ● denote the ratios obtained in this work. A solid line is a linear fit of all values, with the exception of hydrogen.

**Figure 4 molecules-28-03365-f004:**
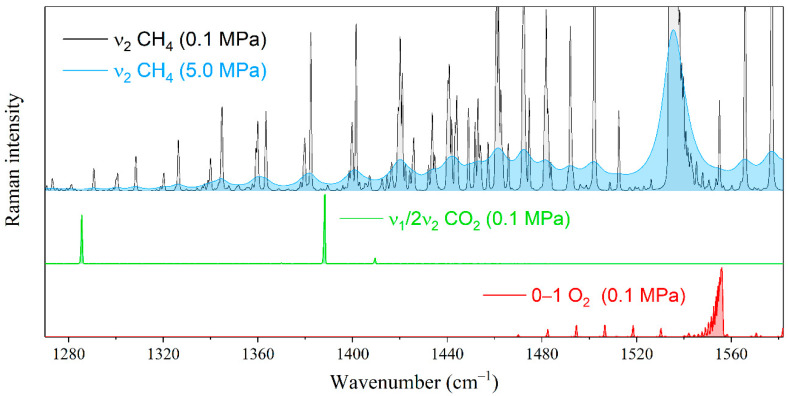
Pressure broadening of the ν_2_ Raman band of methane in the range of characteristic peaks of oxygen and carbon dioxide. The spectral resolution is 0.5 cm^−1^.

**Figure 5 molecules-28-03365-f005:**
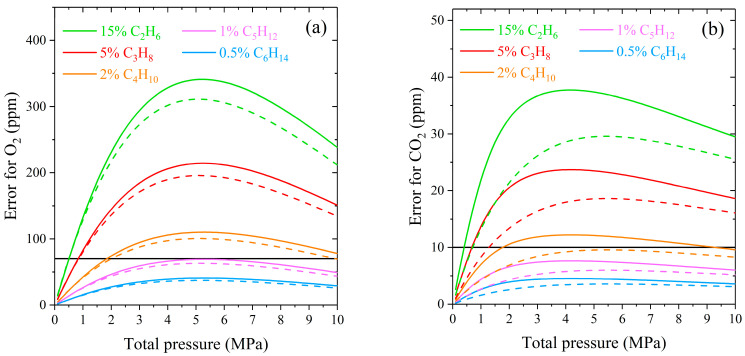
Absolute measurement error of (**a**) O_2_ and (**b**) CO_2_ in the case of neglecting the broadening of the ν_2_ band of methane by alkanes C_2_–C_6_ at the spectral resolution of 0.5 cm^−1^ (solid lines) and 5 cm^−1^ (dashed lines). The solid horizontal lines represent the LOD of O_2_ (70 ppm) and CO_2_ (10 ppm) in natural gas.

**Figure 6 molecules-28-03365-f006:**
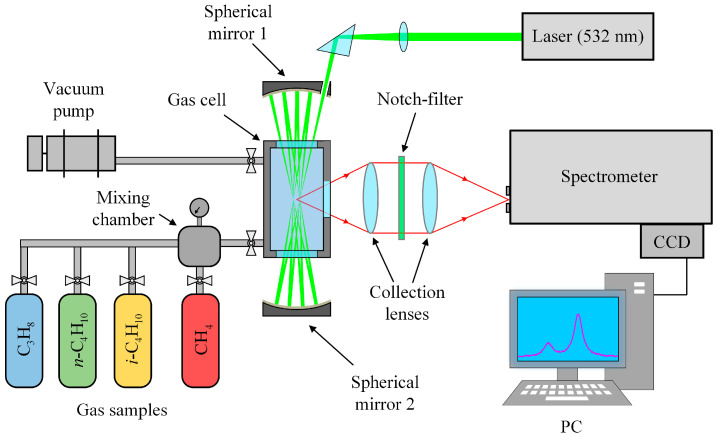
Schematic diagram of the experimental setup (see main text for details).

**Table 1 molecules-28-03365-t001:** Measured broadening coefficients at 298 K for the R- and S-branch transitions in the region of the ν_2_ band of methane perturbed by methane, propane, n-butane, and isobutane.

*J*″	R Branch				S Branch			
	*γ*(CH_4_)	*γ*(C_3_H_8_)	*γ*(*n*-C_4_H_10_)	*γ*(*i*-C_4_H_10_)	*γ*(CH_4_)	*γ*(C_3_H_8_)	*γ*(*n*-C_4_H_10_)	*γ*(*i*-C_4_H_10_)
0	–	–	–	–	81 ± 5	110 ± 10	114 ± 10	122 ± 11
1	77 ± 3	105 ± 7	108 ± 7	116 ± 7	82 ± 2	109 ± 5	113 ± 5	117 ± 5
2	71 ± 3	97 ± 7	100 ± 7	107 ± 7	79 ± 2	108 ± 6	116 ± 6	118 ± 6
3	78 ± 1	104 ± 3	108 ± 3	112 ± 3	76 ± 1	104 ± 3	110 ± 3	111 ± 3
4	71 ± 2	94 ± 4	98 ± 4	101 ± 4	74 ± 1	101 ± 2	104 ± 2	110 ± 3
5	74 ± 2	100 ± 5	108 ± 5	110 ± 5	68 ± 1	93 ± 3	97 ± 3	101 ± 3
6	65 ± 1	88 ± 4	94 ± 4	96 ± 4	63 ± 1	87 ± 2	96 ± 2	97 ± 2
7	66 ± 1	91 ± 3	96 ± 3	97 ± 3	63 ± 1	88 ± 2	93 ± 2	92 ± 2
8	60 ± 2	83 ± 4	87 ± 4	88 ± 4	58 ± 1	81 ± 2	85 ± 2	85 ± 2
9	52 ± 2	72 ± 3	74 ± 3	78 ± 3	53 ± 1	75 ± 3	80 ± 3	84 ± 3
10	61 ± 3	83 ± 5	86 ± 5	91 ± 5	52 ± 2	75 ± 4	80 ± 4	83 ± 4
11	59 ± 4	81 ± 7	84 ± 7	87 ± 7	52 ± 4	74 ± 7	80 ± 8	83 ± 8
12	54 ± 6	74 ± 9	77 ± 10	80 ± 10	–	–	–	–
13	45 ± 6	63 ± 9	69 ± 10	70 ± 10	–	–	–	–

All coefficients are in units of 10^−2^ cm^−1^/MPa. *J*″ denotes the quantum number of the lower rotational state.

**Table 2 molecules-28-03365-t002:** Polarizabilities and ratios of the broadening coefficients of methane lines perturbed by various gases to the self-broadening.

Component	*α*, 10^−3^ nm^3^	$\langle \gamma_{CH_{4}-X}/\gamma_{CH_{4}-CH_{4}}\rangle$		References
		Estimated	Measured	
CH_4_	2.448	1.00	1.00	–
C_2_H_6_	4.226	1.18	1.20	[18]
C_3_H_8_	5.921	1.32	1.38	This work
*n*-C_4_H_10_	8.020	1.48	1.45	This work
*i*-C_4_H_10_	8.009	1.48	1.50	This work
C_5_H_12_	9.879	1.60	–	–
C_6_H_14_	11.630	1.71	–	–
N_2_	1.710	0.77	0.75	[39,46,48,50,74,75,76,77]
O_2_	1.562	0.72	0.71	[39,46,48,49,50,74,75,76]
H_2_	0.787	0.99	0.80	[39,50,75,78]
CO_2_	2.507	0.88	0.91	[51,52,53]
H_2_O	1.501	0.77	0.84	[75]
H_2_S	3.631	1.08	–	–
CO	1.953	0.82	–	–
Ar	1.664	0.72	0.66	[39,46,50,57,74]
He	0.208	0.50	0.51	[39,46,50,51,54,57,74,75]
Ne	0.381	0.35	0.42	[57,74]

## Data Availability

Data are contained within the article.

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
