# Peer review of "Broadening of the ν2 Raman Band of CH4 by C3H8 and C4H10"

_molecules, 2023, doi:10.3390/molecules28083365_

Round 1

Reviewer 1 Report

This article is very well written, and it provides valuable experimental data. Please see the following comments.

  1. The broadening in figure 1 isn’t very obvious and adding an additional re-scaled plot might be helpful for visualizing the broadening.
  2. It would be nice to briefly explain all the bands of methane.   
  3. It would be very helpful to have a schematic diagram of the experimental setup.  
  4. More detailed explanation on how the wavelength is changed to collect the spectra shown in the article is very important.

I don't have further comments on this manuscript. This article is one of the better ones that I reviewed.

Author Response

Reviewer #1.

C1: The broadening in figure 1 isn’t very obvious and adding an additional re-scaled plot might be helpful for visualizing the broadening.

R1: We have changed and re-scaled Figure 1. We hope this will be helpful for visualizing the broadening.

C2: It would be nice to briefly explain all the bands of methane.

R2: We have added a brief explanation to Section 3.1: “The structure of a methane spectrum is quite complex due to the high symmetry of the molecule. The four fundamental modes ν1 (2917 cm–1), ν2 (1533 cm–1), ν3 (3020 cm–1), and ν4 (1311 cm–1) exhibit an approximate relation of stretching and bending vibrations with ν1 ≈ ν3 ≈ 2ν2 ≈ 2ν4 resulting in vibrational levels being grouped into clusters with levels of similar energy, called polyads [64,65]. The first one is a dyad and includes the ν2 and ν4 vibrational modes, covering a wide range of 800–2000 cm–1 at room temperature [30]. The very weak transitions of the normally Raman inactive ν4 band appear in the Raman due to the Coriolis interaction with the active ν2 band [66,67] and can be neglected in the studied spectral range.”

C3: It would be very helpful to have a schematic diagram of the experimental setup.

R3: Done.

C4: More detailed explanation on how the wavelength is changed to collect the spectra shown in the article is very important.

R4: The spectra were obtained simultaneously in the entire range 1505–1825 cm–1 using a non-tunable laser. The scattered light is decomposed using a 2400 lines/mm grating. The decomposed radiation is transformed into a spectrum using a 2048×512 pixel CCD camera. We have updated the text: “Polarized radiation from a 532-nm non-tunable solid-state continuous-wave Nd:YAG laser (SLN-532-5000, Cnilaser) with an output power of 5 W was focused into a metal gas cell to excite scattering. … The simultaneously recorded spectral range was 1505–1825 cm–1 with a resolution of 0.5 cm–1 at 1600 cm–1. The relationship between the CCD pixels and wavenumbers was obtained using a third-order polynomial to transform an intensity distribution of decomposed light into a spectrum.

Reviewer 2 Report

In my view this may be considered for publication after the following modifications:

1.    English in the Manuscript should be thoroughly checked and corrected.

2. why  did authors choose following condition?

 binary mixtures of methane/propane, methane/n-butane,

and methane/isobutane (60/40 mol%) at pressures between 0.1 and 0.44 MPa

3.Is there any other way to compare with the results other than simulation?

  4. The Conclusions section should include:

                              * A highlight of your hypothesis, new concepts and innovations.

                              * A summary of key improvements compared to findings in literature

                              * Your vision for future work

Author Response

Reviewer #2.

C1: English in the Manuscript should be thoroughly checked and corrected.

R1: The English language and style have been improved.

C2: Why did authors choose following condition: binary mixtures of methane/propane, methane/n-butane, and methane/isobutane (60/40 mol%) at pressures between 0.1 and 0.44 MPa?

R2: We have added the following text to the Methods section: “On the one hand, the fraction of a foreign gas in a mixture with methane should be high enough to detect the effect of a foreign broadening of the methane lines. On the other hand, the number density of methane must be high to provide a sufficient Raman signal. According to the criterion proposed by J. Kojima and Q.-V. Nguyen [63], the maximum pressure of a methane mixture when the impact approximation is valid can be estimated as:

,

(1)

where νsep is the averaged line separation between consecutive rotational-vibrational lines of methane, and Γ is the average width (fwhm) of methane lines (was set to 1.4 cm–1/MPa). Using positions of methane lines from Ref. [64], values for νsep and Pmax, averaged over three spin symmetries of methane, were found to be about 0.7 cm–1 and 0.5 MPa, respectively. Thus, we recorded the spectra of binary mixtures of methane/propane, methane/n-butane, and methane/isobutane (60/40 mol%) at pressures between 0.1 and 0.44 MPa to study the broadening processes. We believe that the chosen mole ratio is adequate to provide sufficient signal-to-noise and the measurable effect of C3–C4 alkanes.”

C3: Is there any other way to compare with the results other than simulation?

R3: An experimental spectrum is always a convolution of a molecular line shape function and an instrument response function. In this regard, the simulation/fitting approach is the only way to retrieve reliable and independent broadening parameters.

C4: The Conclusions section should include:

* A highlight of your hypothesis, new concepts and innovations.

* A summary of key improvements compared to findings in literature

* Your vision for future work

R4: We have revised the Conclusions section: “In this work, the averaged broadening coefficients for the R(J) and S(J) manifolds of the ν2 band of methane perturbed by propane and butanes were measured for the first time. It was found that the methane-propane and methane–butanes interactions significantly amplify the collisional broadening of methane lines in comparison with the methane-methane broadening. The C3H8-, n-C4H10-, and i-C4H10-broadening coefficients were found to be 38%, 45%, and 50% higher than the CH4-broadening, respectively. Foreign-gas broadening parameters for the R(1–10) and S(0–10) transitions were measured within a precision of 7% based on uncertainties from mole fraction, gas pressure, signal-to-noise of the recorded spectra, and instrument response function. Further improvement of the measurement accuracy of the broadening coefficients is possible mainly by enhancing the spectral resolution. The effect of foreign-gas broadening will be more pronounced in those spectra where the contribution of the instrument response function will be less. Higher resolution and signal intensity can be achieved by using nonlinear Raman spectroscopy, e.g., coherent anti-Stokes Raman scattering or stimulated inverse Raman scattering. The methane spectrum can be correctly simulated using the line list for methane and obtained parameters. We believe that our data will improve the accuracy of the analysis of methane-bearing gases by Raman spectroscopy.

We have shown that the van der Waals potential, which accounts for only attractive forces, is a suitable approximation to predict the collisional broadening coefficients for rotational-vibrational lines of methane in various media. Within this approximation, the broadening coefficients are proportional to the product of the polarizability of a perturbing molecule and the reduced mass of colliding molecules. We have estimated the averaged C5H12-, C6H12-, CO-, and H2S- broadening coefficients of methane lines from this relationship for the first time. The broadening parameters for J-manifolds can be reconstructed from the distribution of methane self-broadening coefficients by quantum number J and the provided ratios. We suppose that this model is also suitable for other tetrahedral molecules. The necessary broadening coefficients for the active compound can be estimated from the polarizabilities of perturbing molecules or atoms.”